# Deep learning algorithm in detecting intracranial hemorrhages on emergency computed tomographies

Almut Kundisch[1], Alexander Hönning[2], Sven Mutze[3,4], Lutz Kreissl[3], Frederik Spohn[3], Johannes Lemcke[5], Maximilian Sitz[5], Paul Sparenberg[6], Leonie Goelz[3,4]*

1 Center for Emergency Training, BG Klinikum Unfallkrankenhaus Berlin, Berlin, Germany, 2 Center for Clinical Research, BG Klinikum Unfallkrankenhaus Berlin, Berlin, Germany, 3 Department of Radiology and Neuroradiology, BG Klinikum Unfallkrankenhaus Berlin, Berlin, Germany, 4 Institute for Diagnostic Radiology and Neuroradiology, University Medicine Greifswald, Greifswald, Germany, 5 Department of Neurosurgery, BG Klinikum Unfallkrankenhaus Berlin, Berlin, Germany, 6 Department of Neurology, BG Klinikum Unfallkrankenhaus Berlin, Berlin, Germany

* Leonie.Goelz@ukb.de

**Data Availability Statement:** All relevant data are within the paper and its Supporting Information files.

## Abstract

### Background

Highly accurate detection of intracranial hemorrhages (ICH) on head computed tomography (HCT) scans can prove challenging at high-volume centers. This study aimed to determine the number of additional ICHs detected by an artificial intelligence (AI) algorithm and to evaluate reasons for erroneous results at a level I trauma center with teleradiology services.

### Methods

In a retrospective multi-center cohort study, consecutive emergency non-contrast HCT scans were analyzed by a commercially available ICH detection software (AIDOC, Tel Aviv, Israel). Discrepancies between AI analysis and initial radiology report (RR) were reviewed by a blinded neuroradiologist to determine the number of additional ICHs detected and evaluate reasons leading to errors.

### Results

4946 HCT (05/2020-09/2020) from 18 hospitals were included in the analysis. 205 reports (4.1%) were classified as hemorrhages by both radiology report and AI. Out of a total of 162 (3.3%) discrepant reports, 62 were confirmed as hemorrhages by the reference neuroradiologist. 33 ICHs were identified exclusively via RRs. The AI algorithm detected an additional 29 instances of ICH, missed 12.4% of ICH and overcalled 1.9%; RRs missed 10.9% of ICHs and overcalled 0.2%. Many of the ICHs missed by the AI algorithm were located in the subarachnoid space (42.4%) and under the calvaria (48.5%). 85% of ICHs missed by RRs occurred outside of regular working-hours. Calcifications (39.3%), beam-hardening artifacts (18%), tumors (15.7%), and blood vessels (7.9%) were the most common reasons for AI overcalls. ICH size, image quality, and primary examiner experience were not found to be significantly associated with likelihood of incorrect AI results.

**Funding:** The authors received no specific funding for this work.

**Competing interests:** The authors have declared that no competing interests exist.

## Conclusion

Complementing human expertise with AI resulted in a 12.2% increase in ICH detection. The AI algorithm overcalled 1.9% HCT.

## Trial registration

German Clinical Trials Register (DRKS-ID: DRKS00023593).

## Introduction

Level V-III trauma centres are not usually equipped to offer 24/7 neurological care. They often rely on radiological reports and teleradiology to evaluate the severity of neurological conditions and the need for on-site monitoring/treatment or transfer to a specialized center. As teleradiology networks continue to develop, bigger centres receive an ever-increasing stream of diagnostic imaging data of variable quality around the clock. This trend demands improvements in the prioritization and speediness of reporting to ensure prompt treatment in emergent cases [1].

Non-contrast head computed tomography (HCT) scans account for the majority of imaging requests made in teleradiology networks [2]. The majority of these HCTs is conducted on weekends between 8 am and 4 pm [2]. 24-hour in-house radiology coverage is far from the norm for many radiology departments [3]. Overnight neuroradiology coverage is even less common, the interpretation of cranial imaging primarily falling to radiology residents or clinicians [4]. A prospective study examining primary radiology reports (RRs) by residents with re-evaluation by a neuroradiologist found a discrepancy rate of 0.6% regarding the presence of intracranial hemorrhages (ICH) [5]. Technical innovations are needed to offset this diagnostic gap during on-call shifts and improve diagnostic rates for subtle findings [6], as only their detection will ensure that both HCTs and patients undergo evaluation by a clinical specialist.

The term artificial intelligence (AI) was coined during the 1940s [7]. Deep learning is a form of machine learning which uses convolutional neuronal networks to solve both simple and complex tasks [8]. In radiology, AI is increasingly perceived as an opportunity to optimize medical care [9]. Multiple AI algorithms for ICH detection have been tested successfully [10–13]. These innovations offer an opportunity to reduce diagnostic errors in high-volume centres at any time of day or night [14, 15].

Typical pitfalls during the interpretation of HCTs for ICH are other hyperdense intracranial structures (such as calcifications), image quality, and the presence of artifacts [16]. While humans learn to differentiate true ICHs through experience, AI analysis software must be trained on specific analysis results. Radiologist and clinicians can improve ICH detection rates by interpreting HCT using coronal and sagittal reconstructions; AI algorithms usually rely solely on axial views [17]. Despite a wealth of articles on AI, it remains unknown whether the use of deep learning algorithms for ICH detection in teleradiology networks poses specific challenges for the AI software itself and/or for the interpretation of results.

According to recent literature, a commercially available, FDA-cleared, and CE-marked triage and notification software (AIDOC) with a sensitivity of 89–95% and a specificity of 94–99% in detecting ICH [2, 10, 18] might be used to increase the detection rates for ICH [19].

The study reported here sought to determine the number of additional intracranial hemorrhages detected by an AI analysis software and to evaluate possible reasons for errors at a level I trauma center with teleradiology network.

## Materials and methods

This retrospective multi-center cohort-study was prospectively registered on the German Clinical Trials Register (DRKS-ID: DRKS00023593) and conducted in accordance with the Declaration of Helsinki 2013. The institutional review board (Medical Association of Berlin, Germany, Eth-46/20) approved the study protocol and waived the need for written consent. The study comprised 7 phases: screening/enrolment; report classification; AI analysis; discrepancy review; endpoint analysis; review of patient records; and statistical analysis. The study protocol, summarized in Fig 1, is in line with the guidance provided by the Strengthening the Reporting of Observational Studies in Epidemiology (STROBE-) Initiative [20].

### Screening and enrolment

We conducted an exploratory retrospective review of consecutive non-contrast HCTs at both the primary study site and 17 teleradiology network hospitals. The review of all HCTs, acquired during 5 months (05/2020–09/2020), took place prior to the routine implementation of a commercially available, FDA-cleared, CE-marked triage and notification software for ICH detection (AIDOC, Tel Aviv, Israel). None of the HCTs had previously been analyzed using

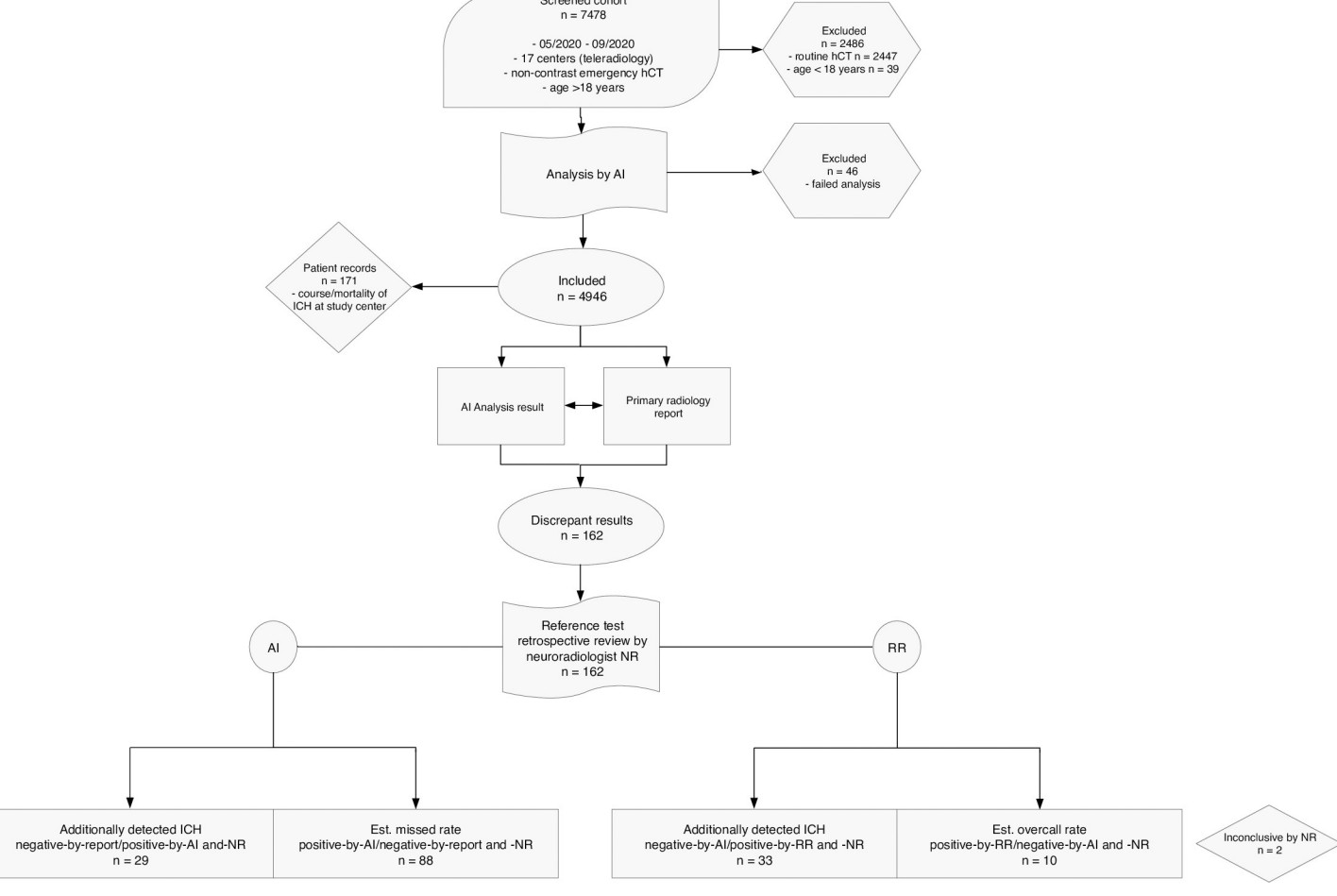

**Fig 1. Study protocol.** Study protocol and conduct followed STROBE-Initiative guidance and describes: patient screening; inclusion and exclusion criteria; comparison of primary RR and AI analysis results with the reference standards, i.e. review by a neuroradiologist (STROBE, Strengthening the Reporting of Observational Studies in Epidemiology; AI, artificial intelligence; RR, radiology report).

the AI algorithm. Scans had a slice thickness of 0.5–5 mm and were performed either in spiral technique with secondary coronal and sagittal reconstructions or in incremental technique. To be eligible, patients had to be aged 18 years or over and had to have undergone examination for an emergent cause such as trauma, neurological deficit, or headache. Routine examinations and contrast-enhanced studies were excluded. All primary RRs were composed by a team of neuroradiologists, radiology consultants, or experienced residents who had been trained at the primary study site.

### Report classification and preparation of cases

After inclusion, RRs were reviewed manually and classified as either "positive-by-report" or "negative-by-report" by an independent radiologist. Additional data collected for each HCT included the number of CT scanner rows, the CT technique (spiral versus incremental), and the experience of the primary examiner (neuroradiologist, radiology consultant, or experienced resident). All HCT were subsequently pseudonymized for analysis by the AI algorithm.

### AI analysis

Prior to this retrospective analysis, a commercially available, cloud-based AI solution for computer-aided triage and prioritization of ICH detection (AIDOC, Tel Aviv, Israel) was selected out of a pool of 11 products to become the first AI tool to be implemented at the teleradiology network. Designed to detect intracranial hemorrhages, it was chosen due to its status as one of the first FDA-cleared products, having received its Section 510(k) clearance in 2018. The algorithm's high level of accuracy has since been described by a number of studies [2, 10, 18, 21].

The AI solution is based on a proprietary two-stage algorithm consisting of a region proposal stage and a false positive reduction stage. The first stage is a 3D Deep Convolutional Neural Network (CNN) trained on HCTs acquired using a diverse range of CT scanners from multiple medical centers around the world. Trained on segmented scans, this network produces a 3D segmentation map from which region proposals are generated and passed as input to the second stage of the algorithm. The second stage then classifies each region as either positive or negative based on features from the last layer of the first stage and traditional image processing methods. Upon detection of suspected positive findings, the AI solution delivers notifications to the radiologist workstation [2].

For our retrospective analysis, axial HCTs were first checked for technical unsuitability for ICH detection (excessive motion artifacts, severe metal artifacts, or an inadequate field of view) by the algorithm. Only HCTs with suspected ICH were flagged (= marked as positive) by the algorithm and the results displayed as color-coded maps on key slices. The independent radiologist received the color-coded maps of flagged cases and classified results into "positive-by-AI" and "negative-by-AI". The color-coded maps marked suspected hemorrhage locations (Fig 2).

### Discrepancy review

Comparison of primary RRs and retrospective AI analyses resulted in two discrepancy categories, namely "negative-by-report/positive-by-AI" and "positive-by-report/negative-by-AI". Discrepant cases were transferred to one of two neuroradiologists (NR) who were blinded to both the RR and AI results and adjudicated on whether or not the HCT in question contained an ICH. Any ICHs detected by the NR were further described according to type (subarachnoid, subdural, epidural, or intracerebral hemorrhage), size, location (supra-/infratentorial, ventricular, lobar), and neighboring structures (calvaria, scull base, falx, or foreign bodies).

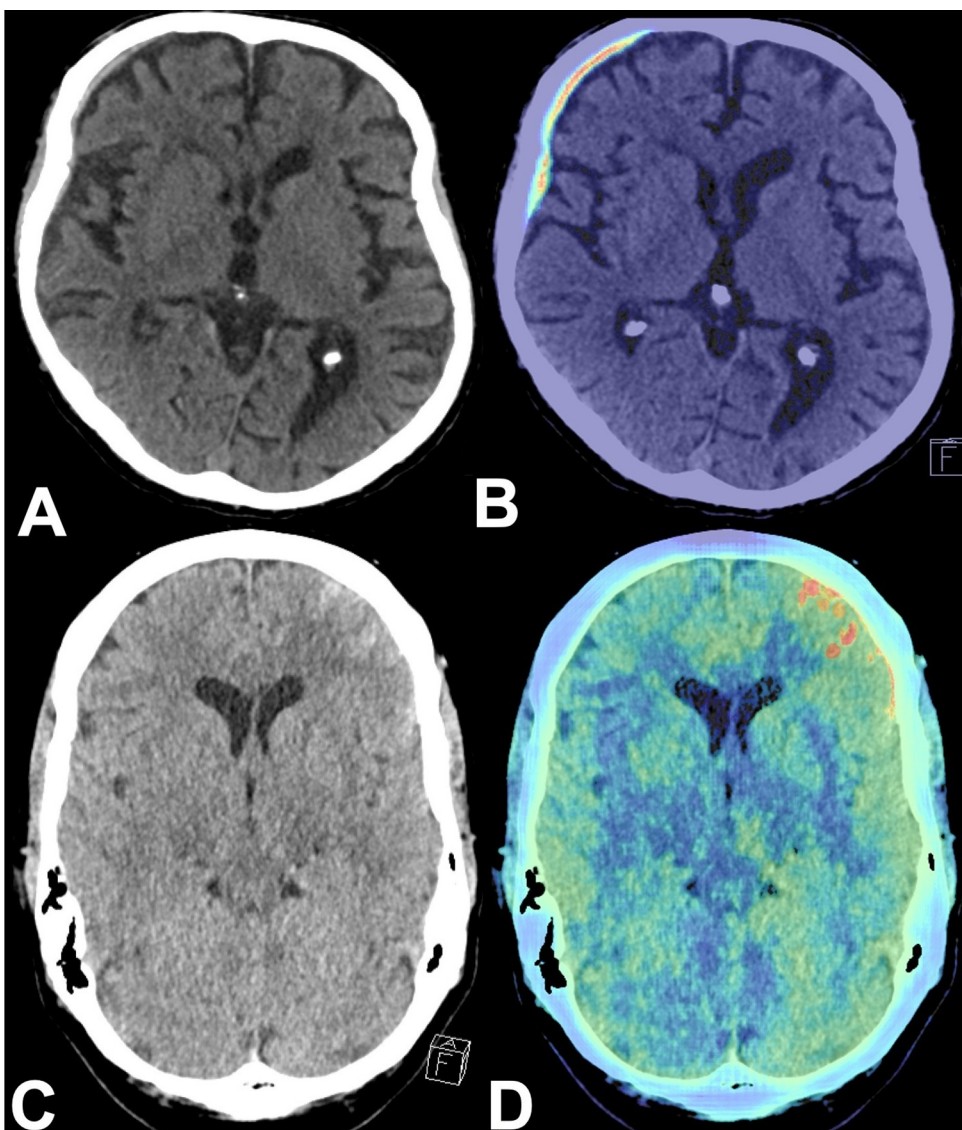

**Fig 2. Color-coded maps.** Axial non-contrast HCT showing a thin SDH of the right frontotemporal hemisphere (A) and a traumatic SAH of the left frontal and temporal lobe (C). B and D: AI analysis results as color-coded maps identifying the relevant findings (ICH) correctly (HCT, head computed tomography, SDH, subdural hematoma; SAH, subarachnoid hemorrhage; AI, artificial intelligence; ICH, intracranial hemorrhage).

### Endpoint analysis

The independent radiologist subsequently determined if the NR agreed with at least one of the findings of the AI analysis ("positive-by-NR") or if she/he disagreed, describing no relevant findings ("negative-by-NR"). Uncertain cases were recorded and discussed by both NRs to determine a consensus result.

The independent radiologist then evaluated the color-coded maps of positive-by-AI/negative-by-NR HCTs for possible underlying reasons, such as: beam-hardening, motion, or metal artifacts; typical hyperdensities (calcifications of falx, plexus, basal ganglia, tentorium, or pineal gland); blood vessel-associated hyperdensities (arteriosclerosis, sinus, developmental anomaly, aneurysm); tumor- and cavernoma-associated hyperdensities; atypical parenchymal

hyperdensities (grey matter or white matter with or without calcifications, basal ganglia without apparent calcifications); and dural patches. The number of additional cases detected by the AI algorithm but missed by the RR corresponded to the negative-by-report/positive-by-AI cases evaluated as positive by the NR; the number of additional ICHs detected exclusively by the RR corresponded to the negative-by-AI/positive-by-report cases evaluated as positive by the NR. Cases overcalled by the RR corresponded to the positive-by-report/negative-by-AI cases evaluated as negative by the NR. Cases overcalled by the AI algorithm corresponded to the positive-by-AI/negative-by-report cases evaluated as negative by the NR. As the study design focused on discrepancies, diagnostic accuracy measures could not be calculated.

## Patient records

Patient records were only accessible at the primary study site via the radiology department. For our final data collection, records of patients with ICH (positive-by-report/positive-by-AI and positive-by-NR) were reviewed for management of ICH (monitoring/conservative in ICU; surgery; angiography; none) and mortality (death in hospital versus discharge). Missed ICHs (negative-by-report/positive-by-AI and positive-by-NR) were examined by two independent neurosurgery attendings who evaluated the following: treatment outcomes in light of the size and location of the ICH; readmission since discharge; anticoagulation medication; and time since HCT. Whether or not there was a medical need to contact a specific patient was determined based on these factors.

## Statistical analysis

All statistical analyses were performed using the SPSS software package for Windows, version 27 (IBM, Armonk, NY, USA). Missing values were not imputed but presented for all relevant variables. Our reporting adhered to the Standards for Reporting of Diagnostic Accuracy statement and recommendations [22]. Descriptive statistics included arithmetic mean, median, standard deviation (SD), minimum and maximum values (range), interquartile range (IQR), as well as absolute numbers (n) and relative proportions (%). We used Pearson's chi squared (two-sided) test to evaluate associations between: (a) ICHs detected/missed by RR and various parameters (size, location, and neighboring structure of ICH; artifacts; CT technique; detector width; location of the study site; experience of radiologist); and (b) ICHs detected/missed ICH by AI and the aforementioned parameters. Where a parameter had multiple ordinal values, these were aggregated into larger categories. P-values of <0.05 were considered statistically significant.

## Results

A total of 7478 non-contrast HCTs acquired between 05/2020 and 09/2020 were screened according to the inclusion criteria. Of these, we excluded 2447 routine HCTs and 39 HCTs of patients aged under 18. Another 46 cases were excluded during the first step of AI analysis due to inadequate image quality (Fig 1).

## Case mix

Out of a total of 4946 consecutive HCTs included in analysis, 2347 (47.5%) were from female and 2596 (52.5%) from male patients. The median age of patients was 72 (IQR 56–83). 2736 (55.3%) cases were sourced from the primary study site (two CT scanners), while the other 2210 (44.7%) were sourced from 17 teleradiology hospitals (17 CT scanners). CT scanners

included one six-row scanner, one eight-row scanner, nine 16-row scanners, five 64-row scanners, two 128-row scanners, and one 2x192-row scanner (S1 Table).

## Primary analysis

A total of 205 (4.1%) reports were classified as hemorrhages by both the RR and the AI, 162 (3.3%) as discrepancies. 62 (1.3%) of the discrepancies were subsequently confirmed as hemorrhages by a NR, resulting in a total of 267 ICHs and an estimated prevalence of 5.4%.

RRs correctly identified a total of 238 (4.8%) ICHs, including 33 (0.67%) cases missed by the AI analysis, resulting in an estimated miss rate of 10.9%. AI analysis flagged a total of 234 ICHs (4.7%), including 29 (0.59%) cases missed by the RR, an estimated miss rate of 12.4%. The AI algorithm identified an additional 12.2% of ICHs which had not initially been detected by RR (Table 1). 88 (1.9%) HCTs which the AI algorithm had flagged as hemorrhages as well as 10 (0.2%) positive RRs were evaluated as incorrect by the NR. Both therefore corresponded to the estimated overcall rates for the AI algorithm and the radiologists. Two cases were classified as inconclusive by the NR.

Neuroradiologists (as primary examiner) missed 4 (estimated miss rate of 10.0%) cases but recorded no overcalls. Radiology consultants missed 19 (estimated miss rate of 11.4%) cases and misclassified 7 as ICH (estimated overcall rate of 0.25%), while residents missed 6 cases (estimated miss rate of 9.8%) and misclassified 3 cases as ICH (estimated overcall rate of 0.34%).

25 (86.2%) of ICHs missed by the RR but detected by the AI algorithm occurred outside of regular working hours, i.e., weekdays between 4:31 pm and 7:30 am or at the weekend.

## Description of incorrect AI/RR results

According to the NRs, the AI algorithm missed a total of 33 ICHs, ranging in size from 1mm to 18 mm (largest diameter). 16 (48.5%) of these misclassifications were associated with hemorrhages smaller than 5 mm in diameter and 17 (51.5%) with hemorrhages 5 mm in diameter or larger. 20 out of 29 (69.0%) ICHs classed by the NR as having been missed by the RR were smaller than 5 mm (S1 Fig). A total of 10 cases (30.3%) of ICH which had been missed by the AI algorithm showed multiple types of hemorrhage, compared with only 3 cases (10.3%) of ICH missed by the RR. Subarachnoid hemorrhages (SAH) were the most common type of missed ICH, accounting for 13 of both the AI (39.4%) and RR (44.8%) evaluations. Subdural hematomas (SDH), epidural hematomas (EDH), intracerebral hematomas (ICB), and intraventricular hemorrhages (IVH) were observed less often (Table 2). ICHs missed by AI were

**Table 1. Summary of main results.**

| Variable | AI analysis | Radiology report |
|---|---|---|
| Correctly classified as ICH | 234 of 267 (87.6%) | 238 of 267 (89.1%) |
| both by AI/RR | 205 (87.6%) | 205 (86.1%) |
| exclusively by AI or RR | 29 (12.4%) | 33 (13.9%) |
| Correctly classified as non-ICH | 4590 of 4679 (98.1%) | 4669 of 4679 (99.8%) |
| both result by AI/RR | 4579 (99.7%) | 4579 (98.1%) |
| exclusively by AI or RR | 11 (0.3%) | 88 (1.9%) |
| Estimated miss rate | 33 of 267 (12.4%) | 29 of 267 (10.9%) |
| Estimated overcall rate | 89 of 4679 (1.9%) | 10 of 4679 (0.2%) |

Results of the primary analysis for the AI algorithm and the primary radiology report.

ICH, intracranial hemorrhage; AI, artificial intelligence; RR, radiology report.

**Table 2. Missed ICH by AI/RR and properties of hemorrhages.**

| Property | Missed AI (n = 33) | Missed RR (n = 29) |
|---|---|---|
| Size of hemorrhage | | |
| <5mm | 16 (48.5%) | 20 (69.0%) |
| > = 5mm | 17 (51.5%) | 8 (27.6%) |
| Missing | 0 | 1 (3.4%) |
| Type of hemorrhage | | |
| Multiple hemorrhages | 10 (30.3%) | 3 (10.3%) |
| SAH | 13 (39.4%) | 13 (44.8%) |
| SDH | 3 (9.1%) | 9 (31.0%) |
| EDH | 3 (9.1%) | 0 |
| ICB | 3 (9.1%) | 2 (6.9%) |
| IVH | 1 (3.0%) | 1 (3.4%) |
| Missing | 0 | 1 (3.4%) |
| Adjacency of hemorrhage | | |
| Under calvaria | 16 (48.5%) | 7 (24.1%) |
| Parafalcine | 3 (9.1%) | 9 (31.0%) |
| Ventricle | 1 (3.0%) | 2 (6.9%) |
| Surrounding foreign body | 1 (3.0%) | 0 |
| Parenchyma | 12 (36.4%) | 10 (34.5%) |
| Level of skull base | 0 | 1 (3.4%) |
| Location of hemorrhages | | |
| Supratentorial | 30 (90.9%) | 23 (79.3%) |
| Infratentorial | 3 (9.1%) | 5 (17.2%) |
| Missing | 0 | 1 (3.4%) |

SAH, subarachnoid hemorrhage; SDH, subdural hemorrhage; EDH, epidural hemorrhage; ICB, intracerebral bleeding; IVH, intraventricular hemorrhage; AI, artificial intelligence; RR, radiology report.

often located immediately under the calvaria (n = 16; 55.2%), whereas ICHs missed by RR were more frequently parafalcine (n = 9; 31.0%) and in the parenchyma (n = 10; 34.5%) (Table 2).

Artifacts were described in 521 (10.9%) of concordant and 83 (51.2%) of discrepant HCT and were most often caused by beam-hardening. They were present in 16 (48.5%) of the ICHs missed by the AI algorithm and in 12 (41.4%) of ICHs missed by the RR.

No statistically significant associations were found between: (a) size, location, or neighboring structure of ICH; artifacts; CT technique; detector width, location of the study site, or experience of radiologist; and (b) the likelihood of an incorrect AI or RR.

## Incorrect AI findings on color-coded maps

In 79 cases (89.8%) positive-by-AI/negative-by-NR results were accounted for by one main reason. Uncommon hyperdensities or calcifications of the parenchyma and dural patches caused 24 (27.3%) (Fig 3), typical hyperdensities caused by calcifications of falx, plexus, basal ganglia, tentorium, or pineal gland accounted for 18 (20.5%) of the false positive AI results (S2 Fig). Beam-hardening artifacts resulted in 16 (18.2%) (S3 Fig), intracranial tumors in 14 (15.9%) (S4 Fig), and unremarkable as well as atypical/pathological blood vessels in 7 (8.0%) (Fig 4) HCTs incorrectly flagged as positive. In 9 (10.2%) cases, false positive AI results were accounted for by multiple findings which contributed equally.

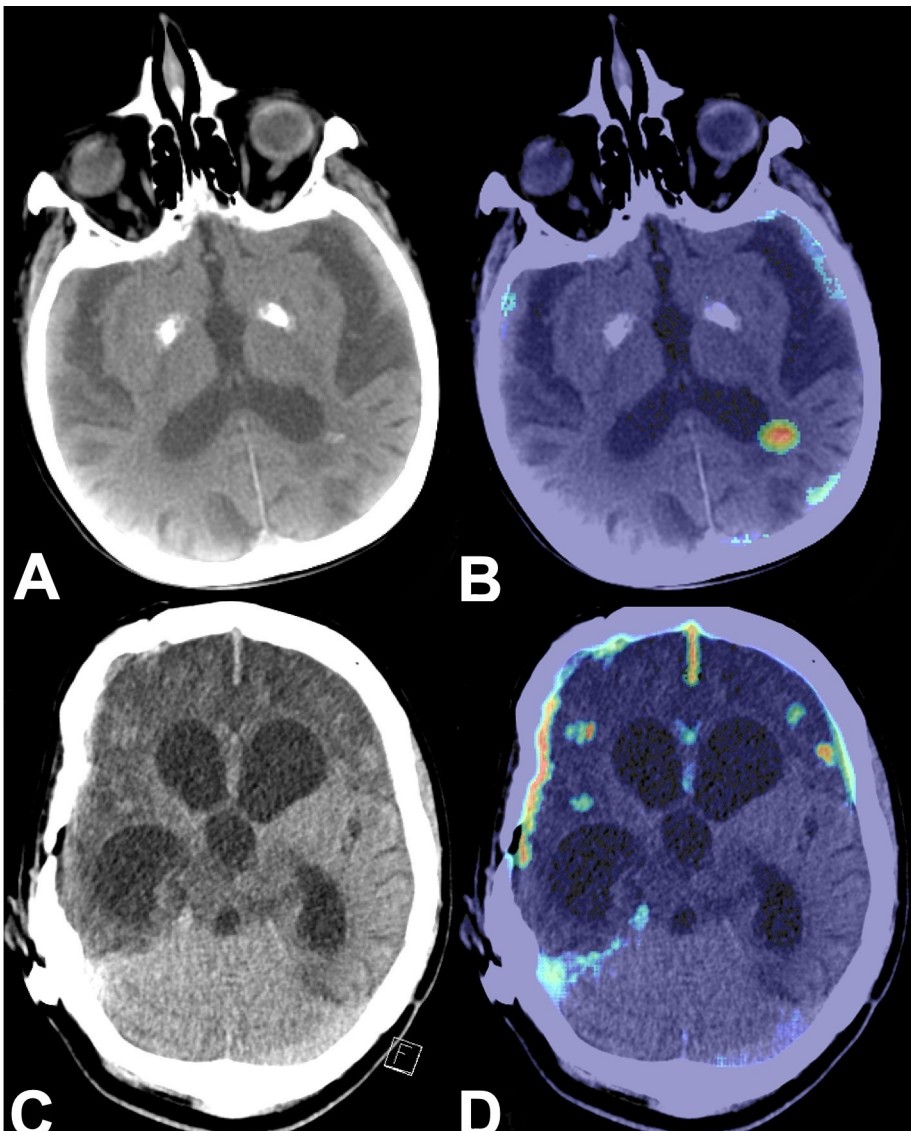

**Fig 3. Uncommon calcifications.** A: Axial HCT showing extremely calcified basal ganglia and calcification of the paraventricular parenchyma of the left dorsal parietal lobe. C: Axial HCT with metal artifact due to a shunt valve. Enlarged ventricles and hypodense parenchyma after non-traumatic SAH. Hyperdense appearance of the frontotemporal dural patch. B and D: Corresponding color-coded maps flagging uncommon calcifications (B), dural patch, falx, tentorium, and residual parenchyma in both the frontal and the right temporal lobes (D) (HCT, head computed tomography; SAH, subarachnoid hemorrhage).

## Relevance, therapy, and mortality

We were able to examine the medical records of 171 patients with confirmed ICH (64.0%) at the primary study site. Of these patients, 97 (56.7%) were monitored in an ICU, 31 (18.1%) underwent surgery, 7 (4.1%) received a cerebral angiography, and 13 (7.6%) were given a poor prognosis. A total of 122 (71.3%) patients were discharged, 29 (17%) patients died in hospital, and in 20 (11.7%) cases, the outcome remained uncertain. The AI algorithm missed the ICH in 3 of the 29 deceased patients (10.3%).

29 cases were initially missed by the RR (negative-by-report/positive-by-NR). Of these, two patients had died due to causes unrelated to the missed ICH. One patient was under followed-

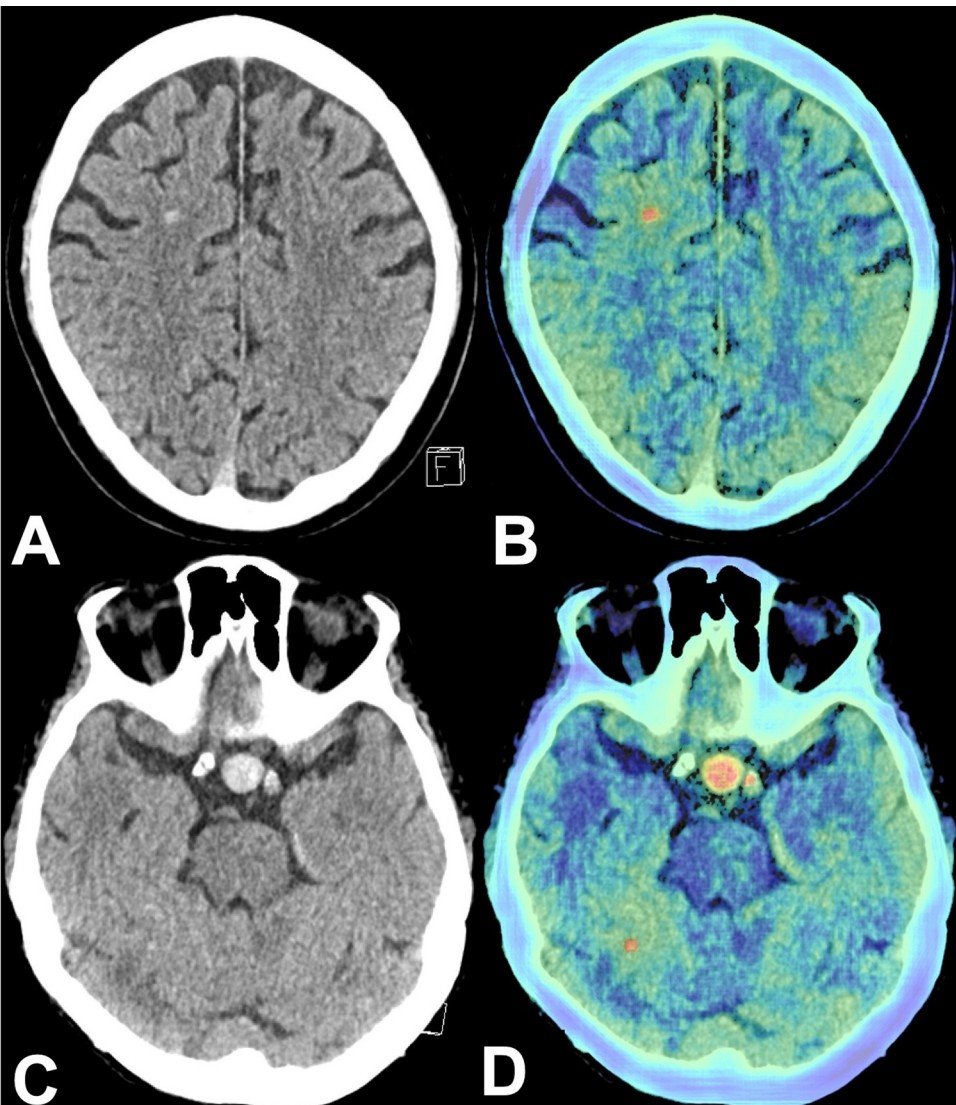

**Fig 4. Atypical and pathological blood vessels.** A: Axial HCT with hyperdense appearance of a DVA of the right frontal lobe. B: Corresponding color-coded map flagging the DVA. C: Axial HCT showing an aneurysm of the left ICA. D: Color-coded map flagging the ICA aneurysm, the left ICA, and a hyperdense spot of the right cerebellar hemisphere (HCT, head computed tomography; DVA, developmental anomaly; ICA, internal carotid artery).

up for a small brainstem cavernoma. Another patient had undergone an MRI during her hospitalization which unmasked her small SAH. Six patients had subsequently received HCTs after repeated falls, which showed resorption of their SAH or SDH. 16 patients whose small SDH or SAH had been missed would usually have been monitored for 1–3 days. Five of these patients were in receipt of anticoagulation medication which would have been paused depending on their comorbidities until a follow-up HCT approximately four weeks later. The same patients might also have undergone repeat HCTs prior to discharge. However, as more than seven months had elapsed, the clinicians agreed that physical and radiological follow-up was no longer medically indicated. Only one young patient with a minor ICH was notified and invited back for a follow-up MRI, a decision prompted by his employer's liability insurance association.

## Discussion

This cohort study confirms that use of AI analysis software can increase the number of ICHs detected in a level I trauma center with teleradiology services. The AI algorithm detected 29 (0.59%) additional ICHs in a cohort of 4946 HCT, a 12.2% increase in the number of detected ICHs. This is in contrast to Rao et al., who reported on the same algorithm retrospectively detecting only 0.24% of missed ICH, most of which were found overlying the convexity and the parafalcine structures [19]. Our analysis produced similar estimated miss rates for both AI analysis and RR. While the likelihood of the AI algorithm missing an ICH was not affected by either ICH size or type, radiologists seemed more likely to miss ICHs of smaller size. One possible conclusion is that the AI algorithm's ability to reliably detect hyperdensities is less dependent on ICH size, while humans are more prone to overlook smaller hyperdensities [23]. It would then follow that combining AI analysis with radiological and clinical experience is likely to benefit patients by highlighting smaller ICH which might otherwise have been missed. Artifacts are another potential risk factor for erroneous AI analysis results. According to our results, AI analysis also missed the majority of ICHs beneath the calvaria and the parafalcine region, most commonly SAH. Difficulties in identifying ICHs in these areas are usually caused by beam-hardening and partial volume artifacts [24]. Artifacts were described in 10.9% of concordant HCTs and in 51.2% of discrepant HCTs. While it appears reasonable that metal and motion artifacts should cause false AI analysis results in any intracranial location, beam-hardening, and partial volume artifacts are common in typical locations of the posterior fossa and close of the skull [25]. The AI algorithm also flagged various typical and uncommon hyperdensities such as tumors, blood vessels, and calcifications as ICHs. The ability to differentiate between hyperdensities and blood or combinations of the two has long been known as difficult to master, and forms part of a long learning-process for residents [16]. Therefore, this skill must be susceptible to errors in machine learning as well and must be afforded special attention during the evaluation of AI analysis results.

Both CT technique and detector width varied greatly in this multicenter study of a teleradiology network, devices ranging from a mobile 6-row scanner HCT to a dual-source 192-row scanner. It is a well-established fact that artifacts can be reduced by modern CT scanners and spiral CT techniques [26–28]. This study did not find any statistically significant associations between: (a) location of the study site (in-house/teleradiology), artifacts, the detector width, or the CT technique; and (b) the likelihood of incorrect AI results. The AI algorithm appears to have operated at a constant level of accuracy despite variations in imaging quality and technique. It might therefore make a valuable addition to teleradiology settings.

Estimated overcall and miss rates for both the AI algorithm and RRs were low, at 1.9%/ 0.2% and 12.4%/10.9% respectively. These results are comparable with previously published sensitivity values of 88.7–95% and specificity values of 94.2–99% [2, 10, 18, 29, 30].

There is a dearth of literature on anatomical reasons for false positive AI analysis and data pertaining to this specific algorithm do not currently exist. Our study identified typical anatomical landmarks which proved difficult for the algorithm. Unusual hyperdensities due to calcifications of the parenchyma, dural patches, and tumors could not be differentiated reliably from ICH. Some typical hyperdensities caused by calcifications of falx, plexus, basal ganglia, tentorium, the pineal gland, and vessels also resulted in AI misclassifications. All of these structures require careful examination by clinicians and radiologists to differentiate overlying blood from misclassification errors by the AI algorithm. Further refinement of the algorithm may reduce its overcall error rate.

Our analysis of patient records points to the relevance of missed ICHs. There was consensus among the clinicians involved that, in 16 out of 29 cases missed by the radiologists, an initial

diagnosis of ICH would have influenced subsequent management. Patients would have been monitored and, wherever possible, anticoagulation medication would have been paused until after a follow-HCT four weeks later. Conversely, the possibility of overdiagnosis by the AI algorithm must also be considered. Some studies have raised concerns regarding the treatment of mild traumatic brain injury, suggesting it might be unnecessary and unresourceful [31, 32]. The majority, however, stress the need for neurological monitoring in patients with small ICHs, which is indicated due to the risk of secondary deterioration, especially in patients on anticoagulants [33–35].

In our study, neuroradiologists were gold standard in HCT interpretation, with low estimated miss and overcall rates. However, 24/7 neuroradiology cover is not the current standard of care and cannot realistically be implemented on a broad basis. Radiological accuracy studies have reported discrepancy rates of 2.3–3.7% between reporting radiologists and neuroradiologists. Discrepancies due to ICH accounted for 0.6% and were found to be mostly the result of missed SDH or SAH in parafalcine and frontal locations [5, 35–37]. There is also the risk that an increase in scheduled CTs and night shifts could reduce the accuracy of radiological reporting [14, 38, 39]. This was exemplified by the results of our study, in which 85% of missed ICHs in this study occurred during on-call shifts. As the majority of emergency HCTs are performed during on-call duty shifts, a prudent approach might be to seek solutions capable of reducing both stress and errors during these sensitive times—particularly given the stark increase in the numbers of emergency HCTs performed during on-call shifts [40–42]. At emergency departments without 24/7 in-house radiology coverage, intelligent solutions for ICH detection may support rapid clinical rapid decision-making and the prompt treatment of critically ill patients.

Algorithm performance can vary with both facility case mix and cohort-specific ICH prevalence [10, 30]. The discrepancy rate between primary RR and AI analysis was 3.3%. Working with AI holds opportunities and challenges for both radiologist and skillful clinicians who, in addition to maintaining their expertise also need to be aware of the pitfalls of the specific software solutions [43, 44]. Specifically, both radiologists and clinicians can and should consult a patient's past and current medical history, previous imaging data, and additional CT reconstructions to form a comprehensive diagnosis where results appear inconclusive (Fig 5 and S5 Fig). When used in combination, AI analysis and human evaluation have the potential to complement their respective strengths and weaknesses. AI solutions should therefore be considered for integration into routine clinical practice as a way to increase the sensitivity of HCT interpretation for ICH while maintaining a low overcall rate [2, 45].

Lastly, it must be stressed that, in many cases with complex conditions (e.g., polytrauma patients with combined intracranial, throracoabdominal, and musculoskeletal injuries), ICH detection only forms a part of a sequence of diagnostic and therapeutic measures. In teleradiology networks, AI analysis could support the prompt and accurate detection of an ICH. Following the identification of concomitant injuries, these patients would then need to be transferred to a level I trauma centers to ensure treatment by an experienced team of radiologists and clinicians [46].

Several limitations of this study must be addressed. Firstly, the retrospective study design is susceptible to selection bias. We addressed this through adherence to the STROBE standards, which enabled us to increase the level of transparency of the selection and inclusion process of consecutive patients. Secondly, given the lack of previous robust studies and sample size calculations, the exploratory nature of this study means it yields mainly descriptive results. Furthermore, while the multicenter approach is one of this study's strengths, it is also associated information or data bias. Access to patient records (for clinical course and mortality data) was limited to the primary study site. Despite this, data from patient records were able to provide

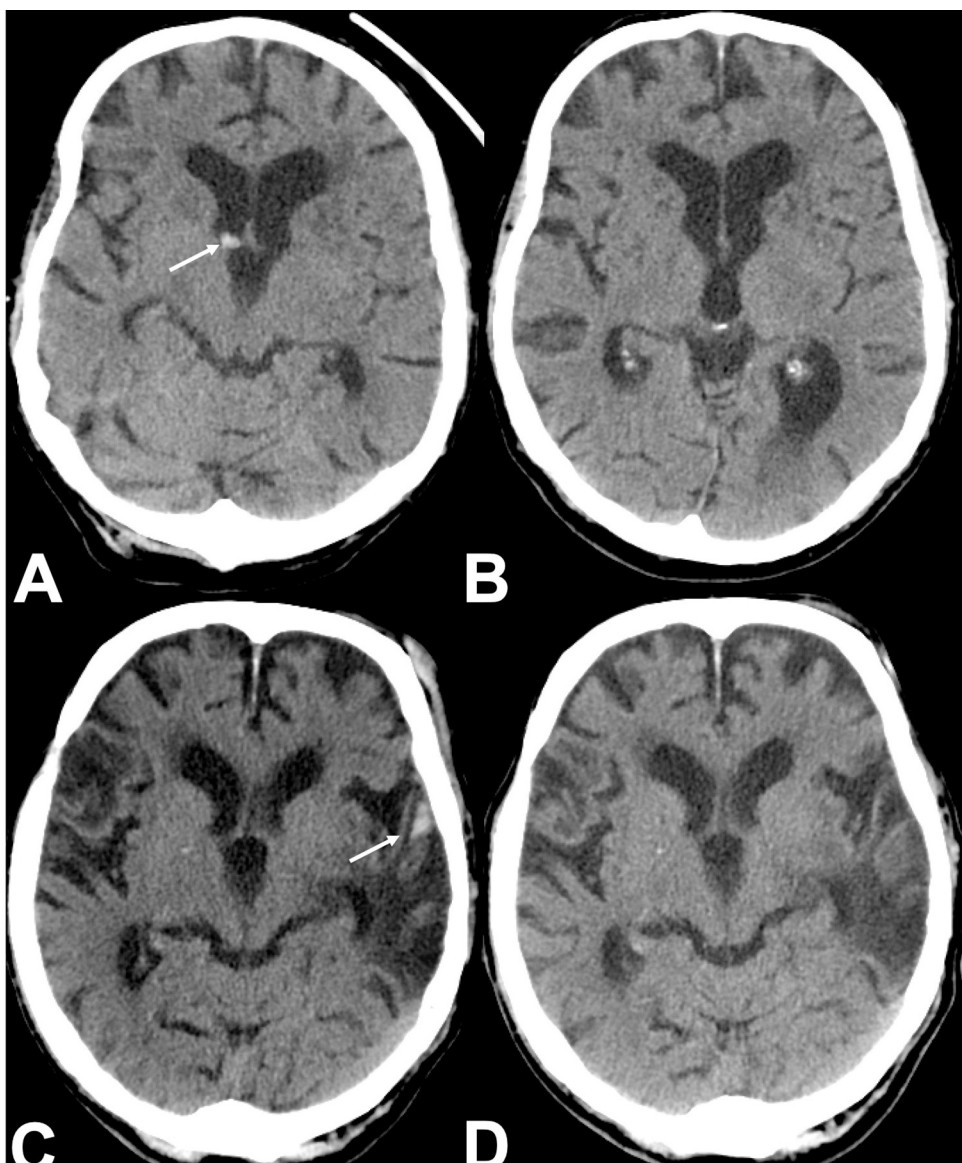

**Fig 5. ICHs missed by the algorithm compared to previous imaging.** Axial HCT showing ICHs (white arrows) missed by the AI algorithm in the right ventricle (A) and the left temporal lobe after ischemic stroke (C). B and D show the corresponding HCTs acquired during the same year, differentiating these hyperdensities from calcifications (HCT, head computed tomography; ICH, intracranial hemorrhage; AI, artificial intelligence).

valuable insight into the potential impact of missed ICH cases. A further limitation was that overcall rates and miss rates could only be estimated. While our study design allowed for the inclusion of a large patient cohort, its focus on discrepancies between initial RR and AI analysis means that it does not permit conclusions regarding diagnostic accuracy measures. However, the likelihood of a HCT with concordant AI and RR findings giving rise to a discordant NR evaluation is vanishingly small. The estimated results can therefore be considered close approximations of the actual miss and overcall rates. Lastly, our study describes the performance of one particular AI algorithm in clinical practice. Additional larger studies are warranted to compare the performance of the available products prospectively [21]. A broader

understanding and the availability of generalized data may aid the institutional decision-making process involved in the procurement and implementation new AI-based solutions.

## Conclusion

The AI algorithm identified an additional 12.2% ICHs. 1.9% of HCTs were overcalled by the AI algorithm; this was often caused by calcifications. ICHs missed by the AI algorithm were mainly located in the subarachnoid space or under the calvaria.

In conclusion, combining human, radiological and clinicals expertise with an AI algorithm is a promising strategy for maximizing ICH detection in high-volume centers with teleradiology services, especially during on-call duty. The identifications of additional ICHs enables prompt monitoring or treatment and could potentially reduce the risk of secondary clinical deterioration in these patients.

## Supporting information

**S1 Checklist. TREND statement checklist.**
(PDF)

**S1 Fig. ICH missed by the RR.** A: Axial HCT showing a thin acute SDH and small contusion of the right temporal hemisphere. B: Corresponding color-coded map identifying the ICHs correctly as main findings. These ICHs were missed by the RR (RR, radiology report; HCT, head computed tomography; SDH, subdural hematoma; ICH, intracranial hemorrhage).
(TIF)

**S2 Fig. Typical calcifications.** A, C, E: Axial HCT without ICH, typical false positive results. Color-coded maps flagging a calcified spot of the falx (B), part of the tentorium (D), and part of the infratentorial plexus at the left lateral aperture (F) (HCT, head computed tomography; ICH, intracranial hemorrhage).
(TIF)

**S3 Fig. Beam-hardening artifacts.** A, C: Axial HCT without ICH. B and D: Corresponding color-coded maps with false positive findings underneath the frontal skull due to beam-hardening artifacts (HCT, head computed tomography; ICH, intracranial hemorrhage).
(TIF)

**S4 Fig. Intracranial tumors.** A, C, E, F: Axial HCT showing hyperdense/partially calcified tumors. A: colloid cyst of the third ventricle, E: intra-/extracranial metastasis of the right frontoparietal hemisphere, G: vestibular schwannoma of the left auditory canal with extra-/intracanicular growth. B, C, G: Color-coded maps flagging the tumors (HCT, head computed tomography).
(TIF)

**S5 Fig. ICH missed by the AI algorithm on three planes.** Axial (A), coronal (B), and sagittal (C) reconstructions showing a thin SAH of the left cerebellar hemisphere (white arrows) which was not flagged by the algorithm (SAH, subarachnoid hemorrhage).
(TIF)

**S1 Table. Characteristics of included cases and scanning environment.** Characteristics of the whole case mix divided into cases with and without intracranial hemorrhage according to the neuroradiologist. IQR, interquartile range; ICH, intracranial hemorrhage; CT, computed tomography.
(DOCX)

**S1 File. Ethics approval.**
(ZIP)

## Acknowledgments

The authors thank Roni Attali and Alexander Böhmcker, as well as the technical team at AIDOC, for their support in processing the cases through the AI solution.

## Author Contributions

**Conceptualization:** Alexander Hönning, Sven Mutze, Leonie Goelz.

**Data curation:** Almut Kundisch, Frederik Spohn.

**Formal analysis:** Alexander Hönning.

**Investigation:** Almut Kundisch, Lutz Kreissl, Johannes Lemcke, Maximilian Sitz, Paul Sparenberg.

**Methodology:** Alexander Hönning, Leonie Goelz.

**Project administration:** Sven Mutze, Johannes Lemcke, Paul Sparenberg.

**Supervision:** Frederik Spohn, Leonie Goelz.

**Visualization:** Sven Mutze, Frederik Spohn, Leonie Goelz.

**Writing – original draft:** Almut Kundisch, Alexander Hönning, Sven Mutze, Maximilian Sitz, Leonie Goelz.

**Writing – review & editing:** Almut Kundisch, Alexander Hönning, Sven Mutze, Lutz Kreissl, Frederik Spohn, Johannes Lemcke, Maximilian Sitz, Paul Sparenberg, Leonie Goelz.

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
