## [Decision Letter · Decision Letter 0]

6 Oct 2021

PONE-D-21-22710Deep Learning Algorithm in Detecting Intracranial Hemorrhages on Emergency Computed Tomographies: Retrospective Multicenter Cohort-Study of 4946 Patients at a Level 1 Trauma Center with TeleradiologyPLOS ONE

Dear Dr. Goelz,

Thank you for submitting your manuscript to PLOS ONE. After careful consideration, we feel that it has merit but does not fully meet PLOS ONE’s publication criteria as it currently stands. Therefore, we invite you to submit a revised version of the manuscript that addresses the points raised during the review process.

We look forward to receiving your revised manuscript.

Kind regards,

Alfio Spina, M.D.

Academic Editor

PLOS ONE

Journal Requirements:

Reviewers' comments:

Reviewer's Responses to Questions

**Comments to the Author**

1. Is the manuscript technically sound, and do the data support the conclusions?

Reviewer #1: Yes

Reviewer #2: No

Reviewer #3: Yes

2. Has the statistical analysis been performed appropriately and rigorously? 

Reviewer #1: N/A

Reviewer #2: No

Reviewer #3: Yes

3. Have the authors made all data underlying the findings in their manuscript fully available?

Reviewer #1: Yes

Reviewer #2: No

Reviewer #3: Yes

4. Is the manuscript presented in an intelligible fashion and written in standard English?

Reviewer #1: Yes

Reviewer #2: No

Reviewer #3: Yes

5. Review Comments to the Author

Reviewer #1: the authors analyze an interesting topic and the article is written in good English.

the use of artificial intelligence in detecting brain haemorrhages in emergency departments could reduce stress for healthcare workers and residences. It should also reduce the overcall diagnosis rate as well as the Numbers of inappropriate neurosurgical consultation . Therefore the author considered only brain haemorrhages which are only part of the injury in this context as in trauma and in case of loss of consciousness. Although highly skilled workers are needed to maintain the system. The author has not analyzed this topic.

The RRs were reports from different centers and provided by radiologists with jeopardized experience and skills . For this reason the comparison does not reflect reals superiority in term of detction rate of AI on resisence and/or radiologists.

selection Bias in setting of retrospective study should be also considered .

Reviewer #2: The language needs language polishing throughout the paper.

The first sentence in the abstract does not make sense! Rephrase.

In general the abstract sounds 'robotic'. Please rewrite in a more vivid way.

Included cases were analyzed by the AI algorithm.: This is not sufficient for a scientific paper. What algorithm? Why this one? Are there no alternatives? How selected? Details about the analysis.

Line 159: What is RR analysis?

The main part of the paper is missing: What is the AI algorithm?

Reviewer #3: The aim of this study is to clarify the utility of deep learning algorithm, which is a non-supervised form of artificial intelligence (AI), in detecting intracranial hemorrages (ICH).

The article is well written in a correct English.

As teleradiology networks continue to develop, AI could be a useful tool to screen the images assuring a prompt treatment in emergent cases.

The study found that AI could increase the number of ICH found in a level 1 trauma center with teleradiology. More interestingly the authors state that in more than 50% of the ICH missed by the radiology, initial diagnosis would have influenced the therapeutic regime.

The topic is interesting. AI could improve diagnostic accuracy in the nearest future.

6. PLOS authors have the option to publish the peer review history of their article (what does this mean?). If published, this will include your full peer review and any attached files.

Reviewer #1: No

Reviewer #2: No

Reviewer #3: No

---

## [Author Response · Author response to Decision Letter 0]

22 Oct 2021

See the Response to Reviewers file as .pdf. Thank you!

---

## [Editor Report · Decision Letter 1]

12 Nov 2021

Deep Learning Algorithm in Detecting Intracranial Hemorrhages on Emergency Computed Tomographies: Retrospective Multicenter Cohort-Study of 4946 Patients at a Level I Trauma Center with Teleradiology

PONE-D-21-22710R1

Dear Dr. Goelz,

We’re pleased to inform you that your manuscript has been judged scientifically suitable for publication and will be formally accepted for publication once it meets all outstanding technical requirements.

Kind regards,

Alfio Spina, M.D.

Academic Editor

PLOS ONE

Additional Editor Comments (optional):

The manuscript has been adequately revised according to the previous Reviewers' comments. Accepted.
---

## [Editor Report · Acceptance letter]

16 Nov 2021

PONE-D-21-22710R1 

Deep learning algorithm in detecting intracranial hemorrhages on emergency computed tomographies 

Dear Dr. Goelz:

I'm pleased to inform you that your manuscript has been deemed suitable for publication in PLOS ONE. Congratulations! Your manuscript is now with our production department. 

Kind regards, 

on behalf of

Dr. Alfio Spina 

Academic Editor

PLOS ONE